# Mechanisms of IGF-1-Mediated Regulation of Skeletal Muscle Hypertrophy and Atrophy

**DOI:** 10.3390/cells9091970

**Published:** 2020-08-26

**Authors:** Tadashi Yoshida, Patrice Delafontaine

**Affiliations:** 1Heart and Vascular Institute, John W. Deming Department of Medicine, Tulane University School of Medicine, 1430 Tulane Ave SL-48, New Orleans, LA 70112, USA; 2Department of Physiology, Tulane University School of Medicine, 1430 Tulane Ave, New Orleans, LA 70112, USA

**Keywords:** insulin-like growth factor-1, skeletal muscle, hypertrophy, atrophy, cachexia, muscle regeneration, autophagy

## Abstract

Insulin-like growth factor-1 (IGF-1) is a key growth factor that regulates both anabolic and catabolic pathways in skeletal muscle. IGF-1 increases skeletal muscle protein synthesis via PI3K/Akt/mTOR and PI3K/Akt/GSK3β pathways. PI3K/Akt can also inhibit FoxOs and suppress transcription of E3 ubiquitin ligases that regulate ubiquitin proteasome system (UPS)-mediated protein degradation. Autophagy is likely inhibited by IGF-1 via mTOR and FoxO signaling, although the contribution of autophagy regulation in IGF-1-mediated inhibition of skeletal muscle atrophy remains to be determined. Evidence has suggested that IGF-1/Akt can inhibit muscle atrophy-inducing cytokine and myostatin signaling via inhibition of the NF-κΒ and Smad pathways, respectively. Several miRNAs have been found to regulate IGF-1 signaling in skeletal muscle, and these miRs are likely regulated in different pathological conditions and contribute to the development of muscle atrophy. IGF-1 also potentiates skeletal muscle regeneration via activation of skeletal muscle stem (satellite) cells, which may contribute to muscle hypertrophy and/or inhibit atrophy. Importantly, IGF-1 levels and IGF-1R downstream signaling are suppressed in many chronic disease conditions and likely result in muscle atrophy via the combined effects of altered protein synthesis, UPS activity, autophagy, and muscle regeneration.

## 1. Introduction

Studies in various models in cell culture, animals, and humans have evaluated cytokines and growth factors that can regulate muscle growth. Insulin-like growth factor-1 (IGF-1) is one of the best-characterized growth factors, and it has been shown to modulate muscle size and play a critical role in regulating muscle function. IGF-1 is thought to mediate many of the beneficial outcomes of physical activity [1,2]. In a study analyzing healthy young subjects, circulating IGF-1 levels were negatively associated with body fat, body mass index (BMI), and total cholesterol and positively associated with aerobic fitness and muscular endurance parameters (VO_2_ peak, sit-ups, push-ups, and repetitive squats) [3]. In contrast, lower IGF-1 levels were associated with various pathological conditions including chronic diseases, inflammation, and malnutrition [4,5]. Since skeletal muscle cells, or myofibers, are postmitotic, their size is determined by a balance between synthesis of new proteins and degradation of old proteins. Under physiological conditions, the rates of protein synthesis and degradation are balanced and the myofiber size is maintained. In cachectic conditions, on the contrary, myofiber protein degradation is accelerated and protein synthesis rate is suppressed, resulting in muscle weakness and fatigue. IGF-1 can regulate both protein synthesis and degradation pathways, and changes in IGF-1 signaling in skeletal muscle can greatly affect myofiber size and function. This review summarizes and discusses different aspects of IGF-1-mediated protein synthetic and degradation pathways in skeletal muscle and its potential application to therapies to treat patients with reduced skeletal muscle function. The signaling pathways downstream of IGF-1 discussed in the following sections are summarized in Figure 1.

## 2. Muscle Protein Synthesis and IGF-1 Signaling

One of the most important function of IGF-1 is its regulation of protein synthesis in skeletal muscle and promotion of body growth. Upon binding to IGF-1, IGF-1 receptor (IGF-1R) phosphorylates an intracellular adaptor protein insulin receptor substrate-1 (IRS-1), which recruits and phosphorylates phosphoinositide 3-kinase (PI3K) followed by Akt phosphorylation. The PI3K/Akt pathway plays a critical role in myotube hypertrophy [6,7], and activation of Akt in rat muscle prevents denervation-induced atrophy [8,9]. Mammalian target of rapamycin (mTOR) is a downstream target of Akt, and in mammalian cells mTOR activity is tightly regulated by amino acid availability to the cells. As amino acids are necessary to build proteins, nucleic acid, glucose, and ATP in the body, mTOR activity is highly correlated with the anabolic/catabolic balance. The IGF-1/Akt/mTOR pathway has been shown to be indispensable in promoting muscle hypertrophy [10]. Akt phosphorylates and inhibits tuberous sclerosis 1 and 2 (TSC1/TSC2), resulting in activation of small G protein Ras homolog enriched in brain (Rheb) via its binding to GTP. GTP-bound Rheb activates mTOR complex-1 (mTORC1), resulting in phosphorylation of p70^S6K^, which promotes protein synthesis by activating ribosomal protein S6, a component of the 40S ribosomal subunit. mTORC1 also phosphorylates 4EBP1, leading to its release from the inhibitory complex with the translation initiation factor eIF4E, the cap binding protein, permitting the binding of eIF4E to eIF4G to form the critical translation initiation complex [11]. When animals were treated with mTOR inhibitor rapamycin, phosphorylation of p70^S6K^ and the release of 4EBP1 from eIF4E inhibitory complex were blocked, leading to inhibition of surgical overload-induced muscle hypertrophy [12]. Consistently, the Akt/mTOR pathway was inhibited during disuse (unloading)-induced atrophy and re-activated after reloading.

Besides mTOR, Akt-mediated phosphorylation of glycogen synthase kinase-3β (GSK3β) is another critical downstream pathway of IGF-1. In muscle hypertrophic conditions, GSK3β is phosphorylated and its activity is inhibited, leading to activation of eIF2B and transcriptional activator β-catenin [13,14]. In contrast, GSK3β activity is increased in a dexamethasone (Dex)-induced atrophy model. Local IGF-1 or constitutively active Akt gene transfer inhibited GSK3β, increased β-catenin levels, and prevented muscle atrophy [15].

In summary, IGF-1/Akt controls two protein synthetic pathways via mTORC1 and GSK3β. Although these pathways are decreased in various muscle atrophy conditions [16,17,18,19], the exact relationship and interaction between these pathways in skeletal muscle atrophy and hypertrophy conditions remain to be determined. IGF-1 also affects protein synthesis via myostatin signaling, and the mechanism is discussed below.

## 3. Muscle Protein Synthesis: Myostatin

Myostatin is a member of the transforming growth factor-β (TGF-β) superfamily, is secreted mainly from skeletal muscle, and negatively regulates muscle mass [20]. Myostatin has been found to be upregulated in cancer, heart disease, HIV, and aging, and systemic administration of myostatin caused cachexia in rodents. Studies have identified crosstalk between myostatin and IGF-1 signaling pathways. In cultured myotubes, myostatin inhibited Akt phosphorylation, resulting in decreased protein synthesis and reduced cell size [21,22,23]. In mice deficient in myostatin, total Akt expression was increased together with increased p70^S6K^ levels [21,24]. These data suggest that myostatin and IGF-1 signaling counteract each other. Indeed, IGF-1 treatment of cultured myotubes blocked myostatin-mediated downregulation of Akt and myotube diameter reduction [23]. Accordingly, the hypertrophic effect of IGF-1 was greater in the myostatin null background [25].

Myostatin signaling is mediated by activin type II receptors (ActRIIA and ActRIIB) and activin type I receptors (ALK4 and ALK5), leading to phosphorylation of Smad proteins (Smad2 and -3). Smad2/3 form a complex with Smad4, which is also a co-mediator of the bone morphogenic protein (BMP) signaling pathway. Therefore, when the myostatin signaling is low, Smad4 becomes more available to BMP signaling, leading to muscle hypertrophy [26]. Studies have suggested that IGF-1 and myostatin/Smad pathways cross-talk at different levels. Akt activation downregulated ActRIIB in denervated muscles, and blocked atrophy-inducing effects of constitutively active ALK4 and ALK5 [27]. Studies in cancer cells have demonstrated direct interaction of Akt and Smad3 to sequester Smad3 outside of the nucleus [28], although it remains to be determined whether the same mechanism exists in skeletal muscle. Although the entire picture of Akt-Smad interaction remains to be determined in skeletal muscle, these data suggest that the balance between competing IGF-1, myostatin, and BMP pathways are critical to maintain muscle mass.

## 4. Muscle Protein Degradation: UPS

The ubiquitin-proteasome system (UPS) is a crucial protein degradation system in eukaryotes, and studies have shown its importance in development of muscle atrophy [29,30]. Muscle atrophy F-box (MAFbx)/Atrogin-1 and muscle RING finger 1 (MuRF1) are the best characterized E3 ubiquitin ligases in skeletal muscle that mediate polyubiquitination of proteins and target them to degradation by the 26S proteasome. MAFbx/Atrogin-1 and MuRF-1 are shown to be increased in various muscle atrophy-inducing conditions, including disuse, denervation, inflammation, aging, glucocorticoid increase, high Ang II, and chronic diseases such as cancer, congestive heart failure, chronic kidney disease, chronic obstructive pulmonary disease (COPD), and AIDS [31,32]. Interestingly, studies have suggested that IGF-1 signaling is altered in many of these conditions, and signaling pathways that regulate MAFbx/Atrogin-1 and MuRF1 are in some part overlapping and regulated by IGF-1 signaling (Figure 1).

Various studies have shown that MAFbx/Atrogin-1 and MuRF1 expression is differentially regulated by FoxO and NF-κB pathways. Inhibition of FoxOs prevented MAFbx/Atrogin-1 increase and protected against muscle atrophy. Although FoxO1 activates both MAFbx/Atrogin-1 and MuRF1 expression in cultured myotubes, the ability of FoxO1 to induce MuRF1 expression is independent of its DNA binding [33]. Similarly, Senf et al. found that FoxO3a induced MAFbx/Atrogin-1 expression via promoter activation, whereas MuRF1 activation did not require FoxO3a DNA binding [34]. Transgenic overexpression of activated IκB kinase β (ΙΚΚβ) in skeletal muscle caused profound muscle wasting, with increased expression of MuRF1 but not MAFbx/Atrogin-1 [35]. In addition, promoter analysis revealed that MuRF1 expression is regulated by upstream NF-κB binding sites, but not FoxO sites in disuse atrophy [36].

It is of note that the IGF-1/PI3K/Akt pathway not only activates FoxO, but also NF-κB signaling via several mechanisms including stimulating p65 transactivation and activation of IKKβ [37]. However, it is not clear whether activation of IGF-1 in skeletal muscle alters NF-κΒ activation and MuRF-1 expression. In myofibers, IGF-1 rapidly and strongly reduced Dex-induced Atrogin-1 expression (~80% reduction after 6 h), whereas MuRF-1 mRNA reduction occurred more slowly (~30% reduction after 18 h) [38]. Importantly, changes in overall proteolysis with Dex and IGF-1 correlated tightly with changes in Atrogin-1 mRNA levels, but not with MuRF1. Consistently, systemic Ang II infusion increased both MAFbx/Atrogin-1 and MuRF-1, whereas IGF-1 inhibited expression and promoter activity of MAFbx/Atrogin-1, but not MuRF-1 [39].

Several substrates of MAFbx/Atrogin-1 and MuRF1 have been identified in skeletal muscle. By yeast two-hybrid screening, eukaryotic initiation factor 3 subunit 5 (eIF3-f) was identified as a target substrate of MAFbx/Atrogin-1 [40]. MAFbx/Atrogin-1 increased eIF3-f degradation in myotubes undergoing atrophy in vitro, and overexpression of eIF3-f caused hypertrophy both in vitro and in vivo. Immunoprecipitation of MAFbx/Atrogin-1 followed by LC-MS/MS analysis in myostatin-treated C2C12 myotubes identified desmin and vimentin as other targets of MAFbx/Atrogin-1 [41]. For MuRF1, myosin heavy chain (MYH) was identified as its target in Dex-induced myotube atrophy model [42]. In addition, by comparing the WT and transgenic mice expressing a RING deletion mutant of MuRF1, which binds but cannot ubiquitinate substrates, Cohen et al. found that atrophying muscles showed a loss of myosin-binding protein C (MyBP-C) and myosin light chains 1 and 2 (MyLC1 and MyLC2) from the myofibril, before loss of MYH [43]. MuRF1 also has been shown to associate with titin and stabilize the sarcomeric M-line [44]. Moreover, MuRF1 is suggested to regulate muscle energy metabolism by targeting creatine kinase [45,46]. However, changes in these target substrates in response to IGF-1 have not been determined in muscle atrophy models.

In addition to well-characterized MAFbx/Atrogin-1 and MuRF1, there are other E3 ubiquitin ligases that are involved in skeletal muscle protein breakdown and are potentially regulated by IGF-1. Milan et al. found that a group of ubiquitin ligases were upregulated in denervated or fasted skeletal muscle, and were blunted in FoxO1, -3, and -4 triple knockout mice (FoxO1,3,4^-/-^) [47]. These ubiquitin ligases include muscle ubiquitin ligase of the SCF complex in atrophy-1 (MUSA1), Fbxo31, and Fbxo21 (also known as SMART). FoxO3 overexpression in myotubes was sufficient to induce MUSA1, but not other ubiquitin ligases. FoxO1 and FoxO3 bind to the promoter regions of MUSA1 and SMART, and the FoxO3 deletion completely blunted the induction of SMART, but not other ubiquitin ligases. These data suggest an overlapping and complex regulation of these ubiquitin ligases by FoxO1, -3, and -4, and therefore by IGF-1.

Nedd4 is a HECT domain ubiquitin ligase that is increased in skeletal muscles after denervation [48,49], unloading [48], and COPD [50]. Nedd4-null mice showed a reduction of IGF-1 and insulin signaling, delayed embryonic development, reduced growth and body weight, and neonatal lethality [51]. Furthermore, skeletal muscle-specific Nedd4 null mice were protected against denervation induced muscle atrophy [52].

Trim32 is a tripartite motif ubiquitin ligase that ubiquitinates and degrades the desmin cytoskeleton, thin filament (actin, tropomyosin, and troponins), and Z-band (α-actinin) [53]. Downregulation of Trim32 in hindlimb muscles reduced fasting-induced breakdown of these contractile and cytoskeletal proteins and muscle atrophy. Furthermore, downregulation of Trim32 in skeletal muscle increased PI3K/Akt/FoxO signaling, enhanced glucose uptake, and induced myofiber growth [54].

TNF receptor adaptor protein 6 (TRAF6) is a member of the TRAF family of adaptor proteins, with the unique property to have E3 ubiquitin ligase activity. TRAF6 is upregulated in skeletal muscle after denervation, starvation, and cancer cachexia development [55,56]. Interestingly, the induction of MAFbx/Atrogin-1 and MuRF1 was suppressed in TRAF6 null mice, suggesting that TRAF6 is an upstream regulator of these E3 ubiquitin ligases. Notably, TRAF6 directly ubiquitinates Akt and inhibits its activity [57]. Although the importance of the potential interaction between IGF-1 signaling and MUSA1, SMART, Nedd4, Trim32, and TRAF6 in skeletal muscle hypertrophy and atrophy remains to be determined, IGF-1 signaling pathway components could be novel targets to regulate these E3 ubiquitin ligase activities in skeletal muscle.

## 5. Muscle Protein Degradation: Autophagy

Another major proteolytic pathway in eukaryotic cells is the autophagy-lysosome system. Autophagy plays a critical role in removal of damaged organelles such as mitochondria, peroxisomes, nuclei and ribosomes, as well as in degradation of damaged or misfolded proteins. Another protective role of autophagy is to provide the degraded cellular components as an energy source to cells especially in the face of sustained starvation. Various skeletal muscle diseases that manifest atrophy and dystrophy such as Pompe disease and Danon disease are associated with lowered autophagic activity [58]. In addition, skeletal muscle-specific knockout mice for Atg7, which acts as an E1-like enzyme critical for autophagy regulation, showed profound muscle atrophy and age-dependent decline in muscle force [59].

Autophagy is regulated by two main pathways that overlap with IGF-1 signaling pathways: mTOR-mediated inhibitory phosphorylation of unc51-like kinase-1 (ULK1) and FoxO3-mediated induction of autophagy-related genes. Since IGF-1 activates mTOR (thus, inhibits ULK1) and inhibits FoxO (thus, inhibits autophagy-related gene expression), it is reasonable to assume IGF-1 inhibits autophagy, although some conflicting results have been reported on the relative importance of mTOR and FoxO pathways in regulation of skeletal muscle autophagy. A first group of studies suggested that mTOR-mediated regulation of autophagy only plays a minor role, at least in skeletal muscle. Only a small (10–15%) induction of autophagy was observed after rapamycin (mTOR inhibitor) treatment in cultured myotubes [60], and rapamycin administration or mTOR knockdown did not induce autophagy in skeletal muscle in vivo [61]. In contrast to these findings, skeletal muscle-specific TSC1-deficient mice (TSCmKO), which show sustained activation of mTORC1, developed a late-onset myopathy related to impaired autophagy [62].

Likely independent of mTOR, Akt activation blocked autophagy via inhibition of FoxO3 [60,61]. Blockade of FoxO3 inhibited the starvation-induced autophagy, and these effects are likely mediated by inhibition of FoxO3-mediated transcriptional activation of autophagy-related genes such as LC3, Bnip3, Beclin-1, Atg4, and Atg12 [63]. Interestingly, Zhao et al. showed that constitutively-active FoxO3 increased protein degradation in cultured myotubes, and, surprisingly, approximately 80% of the effect was mediated by autophagy [60].

These data suggest that both IGF-1/Akt/mTOR and IGF-1/Akt/FoxO pathways inhibit autophagy. However, few studies have extensively analyzed the effect of IGF-1 in skeletal muscle autophagy, and conflicting evidence has been presented. Nakashima et al. treated chicken myotubes with IGF-1 and found that LC3-I to LC3-II conversion, a critical step for autophagosome formation, was decreased [64]. In contrast, Ascenzi et al. showed that LC3-I to LC3-II conversion, which is normally decreased during aging, was increased in mice with skeletal muscle-specific overexpression of IGF-1 [65]. To understand these discrepancies, it is important to note that autophagy involves dynamic and complicated processes, and it has been a challenge in autophagy research to capture a dynamic process with static measurements [66]. Neither of the above studies measured the autophagic flux (i.e., dynamic process of autophagy), therefore more studies are required to understand the role of IGF-1 in regulation of autophagic flux in skeletal muscle in vivo. In other cell types, IGF-1 has been shown to inhibit autophagy. In human colorectal carcinoma drug-resistant cells, IGF-1 inhibited autophagy via Akt/mTOR pathway [67]. IGF-1 knockdown increased autophagy via reduction of Akt/mTOR in aged bone marrow mesenchymal stem cells (BM-MSCs) in hypoxic condition and protected cells against hypoxic injury [68]. This IGF-1-mediated autophagy reduction is suggested to be involved in cellular senescence and longevity. Long-term exposure of quiescent human fibroblasts to IGF-1 reduced viability and increased senescent cells, associated with reduced autophagy and dysfunctional mitochondria. These effects were reversed by rapamycin treatment (mTOR inhibition). Consistently, autophagy is increased in mouse fibroblasts in vivo with lowered IGF-1 levels [69].

In various muscle atrophy conditions such as disuse and denervation, autophagy has been shown to be activated [70]. Although IGF-1 has been used in attempts to prevent muscle atrophy in various models, careful evaluation of autophagy is not always conducted. In models such as cancer cachexia, in which UPS-mediated protein breakdown in increased, overall autophagic activity is likely decreased despite the observation of increased autophagy marker such as Beclin-1, p62, and LC3B [71]. Similarly, in the Ang II-induced muscle atrophy model, autophagy is reduced and likely caused accumulation of dysfunctional mitochondria and impaired skeletal muscle energy metabolism [72]. In both of these models, IGF-1 is reduced [73,74] and IGF-1 administration rescued muscle atrophy [39,75,76]. However, in C26 tumor-bearing mice, neither inhibition nor activation of autophagy rescued the muscle function, and both treatments worsened the outcome [77]. The IGF-1 pathway could still be a promising target to treat muscle atrophy where both autophagy and UPS are activated, and protein synthesis is decreased, as IGF-1 activation could theoretically normalize all of these pathways. However, more careful evaluation of IGF-1′s effects on autophagy is necessary for the development of therapies, as both excessive activation and insufficiency of autophagy could be deleterious to skeletal muscle.

## 6. Muscle Energy Homeostasis: AMPK and IGF-1

5′-adenosine monophosphate-activated protein kinase (AMPK) is an intracellular sensor of ATP consumption and acts as a key regulator of skeletal muscle metabolism. When ATP level is low (thus AMP/ATP ratio is high), AMPK is activated and protein synthesis, which consumes ATP, inhibited. Furthermore, activated AMPK promotes ATP-producing catabolic processes including glucose and fat oxidation, UPS- and autophagy-mediated protein degradation [78]. Via these mechanisms, dominant-negative AMPK overexpression in skeletal muscle or skeletal muscle-specific AMPK gene deletion increased muscle mass [79,80,81]. Mechanistically, AMPK targets two major components of IGF-1 signaling: mTOR and FoxO. AMPK decreases protein translation via activation of mTORC1 and promotes protein breakdown via activation of FoxO1 and FoxO3, which in turn increase UPS and autophagy-related genes (Figure 1). Therefore, it is consistent with these mechanisms that pharmacological or genetic activation of AMPK blocked overloading-induced muscle hypertrophy [82,83]. However, the role of AMPK in muscle atrophy is unclear. In rodent muscle unloading-induced atrophy models, both increased and decreased AMPK activity has been reported [84,85,86,87]. In these models, genetic inactivation of AMPK prevented muscle atrophy [88,89]. In contrast, in Ang II-induced muscle atrophy model, AMPK activity is reduced, and pharmacological and genetic AMPK activation restored muscle mass [90,91]. The proposed mechanistic model is that elevated Ang II reduces ATP content in skeletal muscle, which is supposed to activate AMPK, while Ang II inhibits AMPK activation, causing severe ATP depletion and muscle atrophy. It is not clear whether muscle ATP content is altered in unloading muscle atrophy models and muscle atrophy is caused in a similar mechanism. Importantly, IGF-1 level is reduced in both of these atrophying conditions and Akt is inhibited in skeletal muscle, although the role of AMPK (which is known to inhibit Akt/mTOR and activate FoxO [78,92]) in relation to IGF-1 signaling in atrophying conditions is not clear.

## 7. Alternative Splicing of IGF-1 mRNA to Produce a Local Form

In addition to circulating IGF-1 secreted by the liver, peripheral tissues including skeletal muscle produce IGF-1. Interestingly, some studies suggest distinct roles between circulating and local IGF-1. The IGF-1 gene contains six exons that are differentially spliced to generate multiple transcript variants that result in different pre-pro-IGF-1s (Figure 2). Although the different pre-pro-IGF-1s eventually give rise to the same mature 70-amino acid IGF-1 molecule, it has been shown that these variants have different stabilities, binding partners, and activity. The first two exons are mutually exclusive for their use, and each exon has multiple transcription initiation sites, therefore generating different 5′-UTRs and N-terminal signal sequences. Transcripts containing exon 1 or 2 are referred to as Class 1 and 2, respectively. Exons 3 and 4 are used in all the variants and encode the B, C, A, and D domains, which are named based on their similarity to those in insulin. The 3′-end of IGF-1 gene generates three types of mRNAs with different termination codons, polyadenylation sites, and 3′-UTRs. The C-terminus of pre-pro-IGF-1, termed as E-peptide domain, thus has the greatest variability within the entire protein. The E-peptide domain includes part of exon 4 (16 amino acids), with differential inclusion of exon 5 and 6; Ea consists of exon 6 (19 amino acids) and Eb of exon 5 (61 amino acids). Due to alternative splicing, Ec consists of part of exon 5 (16 amino acids) and part of exon 6 (8 amino acids). Note that these are terminologies for human IGF-1; rodents’ equivalent of human Ec is termed as Eb, as they do not express human Eb-equivalent form. Overall, this alternative splicing generates at least 6 pre-pro-IGF-1: Class 1-Ea, Eb, Ec, and Class 2-Ea, Eb, and Ec. Studies have suggested distinct functions among these different forms of proteins. For instance, Class 1 proteins have a longer signal peptide that is potentially myristoylated and may retain the protein in the ER during the translation process, whereas Class 2 mRNAs are highly expressed in the liver, the primary source of circulating IGF-1. Therefore, Class 1 peptides represent a locally-produced autocrine/paracrine form, and Class 2 peptides represent the circulating endocrine form in the body. Bikle et al. demonstrated that muscle atrophy is more pronounced after ablation of muscle IGF-1 production than when liver IGF-1 production is inhibited [93], suggesting that local IGF-1 is a crucial factor for muscle hypertrophy. However, Temmerman et al. demonstrated the deletion of exon 2 (thus Class 2 mRNAs) in mice did not affect viability, growth, and maintenance of circulating IGF-1 levels [94], and the exact physiological roles of Class 1 and Class 2 proteins remain to be determined. For the E peptide domain, Annibalini et al. identified a highly conserved N-glycosylaton site in the Ea domain, which regulated intracellular pro-IGF-1Ea level via prevention of proteasome-mediated degradation and subcellular localization [95]. Interestingly, Durzyńska et al. found that the predominant forms that are expressed in skeletal muscle are pro-IGF-1s, which contain E peptide, rather than mature IGF-1. Both glycosylated and non-glycosylated forms of pro-IGF-1 were expressed in skeletal muscle, whereas non-glycosylated pro-IGF-1 is more potent to activate IGF-1R [96]. Ascenzi et al. analyzed the effects of IGF-1-Ea and IGF-1-Eb in skeletal muscle and found that only IGF-1-Ea promoted a pronounced hypertrophic phenotype in young mice. Interestingly, however, both isoforms of IGF-1 were protective against age-related loss of muscle mass and force [65]. These data suggest that E domains regulate not only IGF-1 production and secretion but also its local activity.

## 8. IGF-1 Binding Proteins in Skeletal Muscle

IGF-1′s actions are regulated by six IGF-1-binding proteins (IGFBPs), which serve as IGF-1 transport proteins. Approximately 98% of IGF-1 exists as a bound form to one of the IGFBPs, with IGFBP3 accounting for 80% of all the binding. The binding of IGF-1 to IGFBPs is either in a binary complex (an IGF-1 and an IBFBP), or a ternary complex consisting of an IGF-1, an IGFBP and an IGF binding protein acid labile subunit (IGFALS). The binding of IGF-1 to IGFBPs and IGFALS significantly prolongs the half-life of IGF-1 in circulation. The half-lives of unbound IGF-1, IGF-1 in a binary complex, and IGF-1 in a ternary complex are less than 10 min, 25 min and more than 16 h, respectively [97,98,99]. Therefore, circulating levels of IGF-1 are greatly affected by IGFBPs and IGFALS. IGFBP3 gene deletion resulted in 40% decrease in serum IGF-1. IGFALS knockout mice showed 60% reduction in serum IGF-1, and also 90% reduction in IGFBP-3 [100]. As IGFBPs bind to IGF-1 with equal or greater affinity compared to IGF-1R, the binding of IGFBPs to IGF-1 is crucial for the regulation of IGF-1′s availability to peripheral tissues. Another important function of IGFBPs is to prevent the potential interaction of IGF-1 with insulin receptor (IR). Since IGF-1R and IR are structurally similar and IGF-1 can bind to IR with lower affinity, IGF-1 could cause hypoglycemic effects if it can freely access to the IR [101,102]. IGFBP3 is expressed in the liver and peripheral tissues, and its hepatic expression is regulated by GH, allowing the coordinated regulation of circulating IGF-1 and IGFBP3 levels. When bound to IGF-1, IGFPB3 blocks its binding to IGF-1R, thereby impairing the downstream signaling. Furthermore, IGFBP3 has been shown to exhibit antiproliferative and proapoptotic actions via an IGF-1/IGF-1R-independent mechanism [103]. Studies suggest different roles of IGFBPs in regulation of skeletal muscle function depending on muscle type, age, and atrophy conditions. In a study analyzing the expression of mouse IGFBPs at different ages [104], IGFBP4 and -5 were found to be increased with age, whereas IGFBP3 and -6 were regulated differently between males and females: IGFBP-3 decreased with age in males but increased in females, while IGFBP-6 decreased with age in females and remained unchanged in males. Transgenic overexpression of IGF-1 did not alter expression of any of the IGFBPs. Huang et al. analyzed two independent datasets of gene profiles in pancreatic tumors, and found that IGFBP3 was dramatically increased in pancreatic ductal adenocarcinoma, which causes cancer cachexia with high prevalence. The conditioned medium from pancreatic cancer cells contained high IGFBP3 and caused significant myofiber wasting, which was prevented by IGFBP3 knockdown or neutralizing antibody [105]. These results indicate that IGFBPs inhibit IGF-1′s action to induce muscle growth and hypertrophy. Consistently, global overexpression of IGFBP5 in mice caused a severe reduction in prenatal and postnatal growth, resulting in increased neonatal mortality and decreased skeletal muscle weight [106]. Similarly, AAV-mediated overexpression of IGFBP2 in skeletal muscle reduced muscle mass and induced a slower muscle phenotype [107]. On the other hand, mice lacking IGFBP3, -4, or -5 developed normally and only IGFBP4 deficient mice showed a modest (85–90% compared to wild type) growth retardation [108], suggesting that other IGFBPs compensate for the loss of IGFBP5. Indeed, triple knockout of IGFBP3, -4, and -5 had significantly smaller body and quadriceps weight (78% and 60% of wild type, respectively). The triple knockout mice showed lower circulating levels of IGF-1 (45% of wild type) and had lower IGF-1 activity measured by IGF-1R phosphorylation in the cells treated with the serum of the animals (37% of wild type). Interestingly, ERK/MAPK phosphorylation was decreased in the skeletal muscle of triple knockout mice, whereas Akt phosphorylation was not altered [108]. Although these studies indicate that IGFBPs inhibit IGF-1 signaling locally, whether or how IGFBPs affect the outcome of IGF-1 signaling, such as protein synthesis, protein degradation, and autophagy, remains to be elucidated.

## 9. Skeletal Muscle-Specific IGF-1/IGF-1R Gene Deletion Studies

Liver is the major source of circulating IGF-1, and liver-specific IGF-1 gene deletion resulted in 70–80% reduction in serum IGF-1 levels [109,110]. These studies showed normal growth of the animals and threw into question the requirement of circulating IGF-1 for postnatal body growth. However, a later genetic study using a mouse strain with conditional liver-specific IGF-1 expression in IGF-1 null background demonstrated that IGF-1 from the liver contributes approximately 30% of the adult body size [111]. These studies indicate that liver-derived circulating IGF-1 certainly plays a significant role in growth of animals, although it cannot explain all of IGF-1′s growth promoting function in the body.

Transgene, AAV, or electroporation-mediated overexpression of a locally-acting isoform of IGF-1 in skeletal muscle increased muscle mass, myofiber cross sectional area (CSA), and maximum isometric force [112,113,114]. These animals were protected against aging-associated loss of muscle mass [113], Dex-induced atrophy [115], and Ang II-induced atrophy [39,76], whereas disuse atrophy was not prevented [116].

To define the roles of growth hormone (GH) and IGF-1 signaling in skeletal muscle, Mavalli et al. treated primary myoblasts with GH and IGF-1 [117]. Utilizing GH receptor (GHR) and IGF-1R deficient myoblasts, the authors found that, although both GH and IGF-1 induced myoblast proliferation and fusion, the effect was primarily mediated by IGF-1. Both skeletal muscle-specific GHR and IGF-1R knockout mice exhibited reduced myofiber size and number, and impaired muscle force, which are associated with diminished myoblast fusion. Interestingly, muscle-specific GHR deficient mice developed marked peripheral adiposity, insulin resistance, and glucose intolerance, none of which were observed in muscle IGF-1R knockout mice. These data suggest that GH’s action to promote muscle development is mainly mediated by IGF-1, whereas GH facilitates normal insulin action in skeletal muscle independently from IGF-1, leading to changes in global nutrient metabolism. While the study by Mavalli et al. used a cre strain driven by the mef-2c-73k promoter, which is active from an embryonic stage, O’Neill et al. generated skeletal muscle-specific IGF-1R-null mice using the skeletal muscle actin promoter, which is active in differentiated muscle cells, and found that these mice did not show altered body weight or muscle mass [118] In the same study, O’Neill et al. generated mice with muscle-specific double knockout of IGF-1R and IR (MIGIRKO). These animals showed a marked decrease in skeletal muscle mass and fiber size and died earlier (between 15 and 25 weeks), likely due to respiratory failure. Surprisingly, however, glucose and insulin tolerance were not affected in MIGIRKO mice, instead these animals showed increased basal glucose uptake in muscle.

## 10. IGF-1, Satellite Cells and Skeletal Muscle Regeneration

Skeletal muscle stem cells, or satellite cells (SCs), are normally quiescent and located between the basal lamina and sarcolemma of the myofiber. During growth and after muscle damage, a myogenic program of SCs is activated, and SCs self-renew to maintain their pool and/or differentiate to form myoblasts and eventually myofibers.

IGF-1 has been shown to increase both proliferation and differentiation of cultured myoblasts [119]. When cells are in the proliferative stage, IGF-1 increased the expression of cell-cycle progression factors, whereas IGF-1 promoted myoblast differentiation when cells are withdrawn from the cell cycle by myogenic regulatory factors such as myogenin. L6E9 cell line is a subclone of the parental rat myoblast cell line L6, and does not express IGF-1 whereas IGF-1R expression is intact. Utilizing these cells, Musaro et al. demonstrated that IGF-1 overexpression in differentiated L6E9 cells resulted in pronounced myotube hypertrophy and myogenin induction [120]. PI3K/Akt and MAPK pathways have been shown to mediate downstream signaling of IGF-1 in these cells, although the relative importance of these pathways seems to differ depending on the model systems analyzed. Blockade of MAPK inhibited IGF-1-mediated L6A1 myoblast (another subclone of rat neonatal myoblast cell line L6) proliferation, whereas blockade of PI3K or mTOR abolished myoblast differentiation [121]. In contrast, SCs isolated from muscle-specific IGF-1 transgenic mice showed enhanced proliferative capacity in vitro, and the effect was mediated by activation of PI3K/Akt, independent of MAPK, and downregulation of the cyclin-dependent kinase inhibitor p27^Kip1^, supporting the role of IGF-1 in regulation of the cell cycle in SCs [122].

In addition to the above-mentioned in vitro studies, a series of in vivo studies have shown the importance of IGF-1 signaling in SC function. Barton-Davis et al. proposed that the increase in skeletal muscle mass and strength in mice that overexpress IGF-1 specifically in skeletal muscle is primarily due to the activation of SCs and increased regeneration [112]. In mice treated with hindlimb gamma-irradiation to prevent SC proliferation, approximately half of IGF-1′s hypertrophic effect was prevented. However, a following study by Heslop et al. showed hindlimb gamma-irradiation does not completely abolish SC function [123], questioning whether the observation in the study by Barton-Davis et al. is due to depletion of SCs. More recent studies presented conflicting evidence whether SCs are required for muscle hypertrophy [124,125], indicating the importance of careful evaluation and selection of appropriate animal model to address the in vivo contribution of SCs to muscle hypertrophy and the role of IGF-1.

Another consideration needs to be given when analyzing IGF-1′s role in SCs is the potential isoform-specific effects of IGF-1. By differential screening, IGF-1 mRNA with the Ec form of E peptide domain (see Section 7) was identified as the transcript that is increased in exercised muscle compared to the resting state, and named mechano-growth factor (MGF). MGF has been shown to stimulate SCs to re-enter the cell cycle and proliferate, facilitating new myofibers to replace damaged myofibers [126] In addition, impairments of IGF-1 splicing to produce MGF were observed during muscle wasting and age-related decline of muscle regeneration [127,128,129]. Attention needs to be drawn to the usage of the MGF terminology, as some studies use it in referring to the Ec portion of the peptide alone, not including the IGF-1 mature peptide (to avoid any confusion, it is called the Ec peptide in this article). Yang et al. showed that, unlike mature IGF-1, the Ec peptide inhibited C2C12 myoblast terminal differentiation, while increasing proliferation in IGF-1R-independent manner [130]. Furthermore, the Ec peptide increased the proliferative lifespan and delayed senescence of SCs isolated from healthy human subjects [131], and increased the number of primary cultured muscle progenitor cells isolated from patients with muscular dystrophies (CMD, FSHD) and amyotrophic lateral sclerosis (ALS) [132]. However, a contradictory study has been reported [133], in which investigators failed to show any effect of the Ec peptide on C2C12 or primary human myoblasts. A study investigating another IGF-1 isoform class 2 IGF-1-Ea showed that this isoform exerts its hypertrophic effect only when the muscles are in growing status (e.g., during postnatal development or during regeneration) [130]. These studies suggest that IGF-1′s effects on SCs differ between isoforms, but no study has been conducted to compare the isoform-specific effects of IGF-1 on muscle regeneration and atrophy in vivo.

IGF-1 seems to regulate SCs in concert with other myogenic factors. The morphogenic factor sonic Hedgehog (Shh) has been reported to be expressed in adult myoblasts and to promote their proliferation and differentiation [134,135]. Both Shh and IGF-1 enhanced Akt and MAPK phosphorylation and myogenic factor expression levels in C2 myoblasts in a dose-responsive manner, having additive effects. In cultured myoblasts isolated from mice with a muscle-specific knockout of Smoothened (Smo), a component of the Shh receptor, IGF-1-induced Akt and MAPK phosphorylation and myogenic differentiation were significantly blocked. Interestingly, Smo physically associates with the IGF-1R, the p85 regulatory subunit of PI3K, and IRS1 in a Shh and IGF-1 dose-responsive manner, indicating that mutual regulation of Shh and IGF-1 occurs at the receptor complex level [136].

Another potential mechanism whereby IGF-1 affects SC function is via regulation of autophagy. Zecchini et al. showed that autophagy is required for neonatal myogenesis and muscle development [137]. Atg7 is an E1-like activating enzyme that regulates fusion of peroxisomal and vacuolar membranes during autophagy, and Atg7 knockdown in SCs caused severe reduction in neonatal myogenesis. Interestingly, the expression of GHR and IGF-1 were reduced in the skeletal muscle of these animals. In primary cultures of neonatal SCs, the defective autophagy decreased proliferation and differentiation, and GH’s action to promote myotube growth was completely abolished. As discussed in Section 5, IGF-1 likely reduces autophagy in skeletal muscle. In addition, IGF-1 is known to be reduced in various muscle atrophy conditions. However, it is not clear whether reduced IGF-1 results in an increased autophagy in these conditions, or whether altered autophagy affects the SC functions in these atrophy conditions.

## 11. Atrophy-Related miRs and Their Potential Regulation of IGF-1 Signaling

Various non-coding RNAs have been proposed to regulate IGF-1 signaling in skeletal muscle. Using miRNA arrays, Li et al. found miR-29b as the only miRNA whose expression was increased in five different in vivo murine muscle atrophy models (denervation, Dex-treatment, fasting, cancer cachexia, and aging) as well as three in vitro atrophy-inducing cell culture models (C2C12 myotubes treated with dexamethasone, TNF-α, and H_2_O_2_) [138]. miR-29b overexpression promoted muscle atrophy, while miR-29b inhibition prevented denervation-induced muscle atrophy. Importantly, the authors found that miR-29b targets two members of the IGF-1/Akt/mTOR pathway, IGF-1 and PI3K (p85α). miR-29b agomir decreased Akt activity and activated FoxO3A, as well as decreased mTORC1 and p70^S6K^ both in vitro and in vivo. However, conflicting evidence was presented by Goodman et al., showing that Smad3 gene transfer to skeletal muscle decreased miR-29 promoter activity, whereas Akt/mTOR activity was decreased and skeletal muscle atrophy was induced [139]. Furthermore, in a mouse model of CKD-induced muscle atrophy, miR-29 was decreased in skeletal muscle [140], and exosome-mediated miR-29 transfer prevented muscle atrophy [141]. In these studies, phosphatase and tensin homolog (PTEN), which suppresses IGF-1 pathway, and transcriptional repressor Yin Yang 1, which suppresses IGF-1 transcription [142], were shown to be targets of miR-29. It is not clear the potential reasons of these discrepancies, although it is interesting that multiple IGF-1 signaling pathway molecules are potentially targeted by one miRNA, and studies are required to investigate its relationship with other miRNAs discussed below.

During myogenesis, the expression of miR-1 and miR-133 are greatly induced [143], whereas these miRs are reduced during muscle hypertrophy [144]. In C2C12 myoblasts, miR-1 and miR-133 are shown to inhibit the IGF-1 pathway by targeting IGF-1, IGF-1R and HSP70 [145,146,147], although their roles in skeletal muscle remain to be determined.

miR-128a is highly expressed in brain and skeletal muscle, and it has been shown to target IRS1 [148]. Inhibition of miR-128a in C2C12 myotubes increased IRS1 protein and Akt activity, resulting in increased the size of the myotubes. Furthermore, administration of antisense miR-128a caused skeletal muscle hypertrophy in mice.

miR-486 is encoded in the intron of the Ank1.5 gene, which functions to connect sarcomeres to the sarcoplasmic reticulum [149,150], and is co-expressed with Ank1.5 mRNA [151] miR-486 is found to target PTEN and FoxO1. PTEN dephosphorylates PIP3, and thus inhibits PI3K’s activity to phosphorylate PIP2 to produce PIP3, resulting in inhibition of Akt. It is suggested that myostatin inhibits miR-486; overexpression of miR-486 induced myotube hypertrophy via activation of Akt [152] and restored Akt activity and muscle mass in CKD-induced muscle atrophy model [153].

Long noncoding RNAs (lncRNAs) are novel class of regulatory RNAs, which are involved in numerous biological processes via interaction with mRNAs and miRNAs, such as miRNA and lncRNA competition for the same mRNA target, and lncRNAs acting as decoys (or sponges) for miRNAs [154]. By RNA sequencing of hypertrophic and leaner broilers, Li et al. identified a novel lncRNA, termed lncIRS1, is upregulated in hypertrophic muscles. LncIRS1 promoted proliferation and differentiation of myoblasts in vitro, and muscle mass and myofiber size in vivo [155]. Mechanistically, lncIRS1 acts as a molecular sponge for miR-15a, miR-15b-5p, and miR-15c-p, all of which interact with IRS1 mRNA. Increased lncIRS1 inhibits the activity of these miRs, leading to activation of IRS1 and muscle hypertrophy.

These studies strongly suggest the involvement of different miRs in IGF-1-mediated hypertrophy and atrophy prevention. However, some conflicting studies have been published as in the case of miR-29, and further studies are required analyzing these miRs in specific hypertrophy and atrophy models, especially in human patients.

## 12. IGF-1 Changes in Chronic Conditions and Aging-Associated Sarcopenia

Local overexpression of IGF-1 has successfully rescued muscles in various chronic and experimental muscle atrophy models including Dex injected rats [115], age-related muscle atrophy [113], hindlimb suspension [156], and Ang II infusion in rodents [39,74,76], as well as in the mouse models of ALS [157] and muscular dystrophy [158,159,160]. Rheumatoid arthritis (RA) is associated with low muscle mass and density, and skeletal muscle of RA patients have been shown to have lower levels of IGF-1, which were associated with the severity of the disease, low appendicular lean mass, and lower myofiber CSA [161]. In a rat RA model, both circulating and skeletal muscle IGF-1 were decreased, the animals showed lower muscle mass, and subcutaneous injection of IGF-1 (100 µg/kg; twice daily for 12 days) increased body and hindlimb muscle weight without changing arthritis. RA increased skeletal muscle MAFbx/Atrogin-1, MuRF1, IGFBP3, and IGFBP5 expression, and IGF-1 treatment attenuated the increase of MAFbx/Atrogin-1, MuRF1, and IGFBP3, but not IGFBP5 [162]. Although a decrease in circulating and skeletal muscle IGF-1 has been reported in various chronic conditions, including cancer, congestive heart failure, chronic kidney disease, and COPD, and aging [5,163] (Figure 2), more studies are required to determine whether IGF-1 administration could be a therapeutic approach to treat muscle atrophy in these patients (discussed below).

In rats bearing AH-130 hepatomas, IGF-1 mRNA expression in hindlimb muscles progressively decreased, whereas that of IGF-1R and IR increased [73]. Circulating and hepatic IGF-1 levels were also decreased in this model, and these changes were associated with increased MAFbx/Atrogin-1 and MuRF1 expression in skeletal muscle. In the Apc^Min/+^ mice, a model of colorectal cancer that develops cachexia, muscle IGF-1 mRNA expression was decreased with suppressed mTOR targets [164]. Similar results were observed in humans, as muscle IGF-1 mRNA was decreased in gastric cancer patients [165]. Interestingly, the reduction of IGF-1 was observed irrespective of the weight loss, suggesting that IGF-1 downregulation precedes cachexia development. In the rat AH-130 hepatoma model, subcutaneous injection of IGF-1 for 16 days attenuated the loss of lean mass at low-dose (0.3 mg/kg/day) and high-dose (3 mg/kg/day), with improvement of spontaneous activity, food intake, and mortality at low-dose treatment [75]. However, in the same animal model, the parenteral administration of IGF-1 did not alter E3 ubiquitin ligase expression or muscle atrophy [73]. The same group of authors also found that phosphorylation of Akt was comparable or increased in skeletal muscle of mice bearing AH-130 hepatomas or C26 colon adenocarcinomas, with hyperphosphorylation of GSK3β, p70^S6K^, and FoxO1 and reduced eIF2α phosphorylation. Electroporation-mediated IGF-1 gene transfer to the hindlimbs of these animals did not alter myofiber size and muscle mass [166]. These data suggest that IGF-1′s effect to treat cancer-induced muscle atrophy may depend on the cancer type, animal species, and/or the route and dose of administration.

Low circulating IGF-1 levels have been associated with an increased risk and worse prognosis of cardiovascular diseases in human patients [167,168,169]. Deficiency in liver-derived IGF-1 caused impaired contractility of cardiac myocytes and compensatory hypertrophic response [170,171]. Importantly, skeletal muscle atrophy is a hallmark of rodent myocardial infarction models of congestive heart failure. In skeletal muscles of these animals, Akt/mTOR/p70^S6K^ signaling is decreased and MAFbx/Atrogin-1 is increased [76,172], and transgenic overexpression of IGF-1 inhibited muscle atrophy [167]. Interestingly, skeletal muscle-specific Akt activation decreased cardiac myocyte hypertrophy, decreased interstitial fibrosis, and restored contractile function in the heart, suggesting skeletal muscle to cardiac communication [173].

In experimental models of CKD, UPS-mediated protein degradation is increased with impaired insulin and IGF-1 signaling [174,175]. Interestingly, SC proliferation and differentiation are impaired in CKD mouse model, and Akt activity was decreased. Kido et al. proposed that advanced glycation end-products (AGEs), which is accumulated in patients with CKD, increases fibroblast growth factor 23 (FGF23) and its receptor Klotho-mediated suppression of insulin/IGF-1, leading to inhibition of S differentiation. In addition to UPS activation and SC inhibition, CKD was associated with increased autophagy in skeletal muscles of human CKD patients [176]. These pathways can be initiated by complications associated with CKD, such as metabolic acidosis, defective insulin and IGF-1 signaling, inflammation, increased angiotensin II levels, abnormal appetite regulation, and impaired microRNA responses [175]. Whether IGF-1 administration can rescue CKD-mediated skeletal muscle atrophy remains to be determined.

Muscle dysfunction is one of the most relevant systemic manifestations of patients with COPD, and lower limb muscle atrophy is frequently observed in COPD patients [177]. Survival in patients with COPD is negatively associated with skeletal muscle dysfunction and lower mass, and COPD exacerbations rapidly induce loss of muscle mass and function [178,179,180,181]. As in other cases of muscle atrophy, UPS-mediated protein degradation is activated in COPD skeletal muscles [50,182]. In addition, SC senescence and reduced regenerative capacity were reported in SCs isolated from COPD patients [183,184], suggesting the lower SC funciton contributes to muscle atrophy in COPD. However, the contribution of IGF-1 in COPD patients is not clear. Circulating levels of IGF-1 were reported to be unchanged in COPD patients [185], and in cachectic vs. non-cachectic patients with COPD [186]. However, IGF-1 levels were decreased during periods of acute exacerbation [187], which is known to result in muscle atrophy. More careful evaluation of IGF-1 levels and signaling is necessary for these patients.

Aging-associated decline in skeletal muscle mass, quality, and strength mostly occurs in type 2 (fast-twitch) muscle fibers and is associated with marked infiltration of fibrous and adipose tissues in the muscle [188]. Both circulating and local IGF-1 levels are reduced in aging [189], with decreased Akt/mTOR/p70^S6K^ in skeletal muscle [9,189,190]. AAV-mediated IGF-1 gene transfer prevented aging-related muscle changes in old mice [191], and, conversely, deletion of liver-specific IGF-1 at one year of age dramatically impaired health span of the mice [192]. Furthermore, the age-related reduction in IGF-1 levels are accompanied by increased IGFBP levels, further decreasing IGF-1 availability to peripheral tissues. Contrary to these findings, Sandri et al. reported only modest to no changes in IGF-1/Akt/mTOR pathway in old human subjects [193].

## 13. Conclusions

We review the role of IGF-1 and its downstream signaling in skeletal muscle atrophy associated with various chronic diseases and aging. IGF-1 regulates skeletal muscle protein synthesis and protein degradation via the UPS and autophagy, and multiple pathways and mechanisms have been identified (Figure 1). IGF-1 has also been shown to activate satellite cell proliferation, although the involvement of these cells in atrophy development in in vivo animal models and human patients remains to be elucidated. One of the difficulties in IGF-1 research in skeletal muscle is that IGF-1 regulates numerous biological pathways, and these pathways likely interact with each other. For instance, growing evidence suggests the involvement of different miRNAs in IGF-1 signaling, and, considering that each miRNA can target multiple mRNAs, careful examination of changes and biological functions of these miRNAs will be required. Furthermore, it is possible that delivering a specific isoform of IGF-1 may be required to have effective activation of downstream signaling. The role and relative importance of IGF-1 signaling likely differs between muscle atrophy models, and further studies are required to develop effective strategies to apply IGF-1 to treat muscle atrophy in human patients.

## Figures and Tables

**Figure 1 cells-09-01970-f001:**
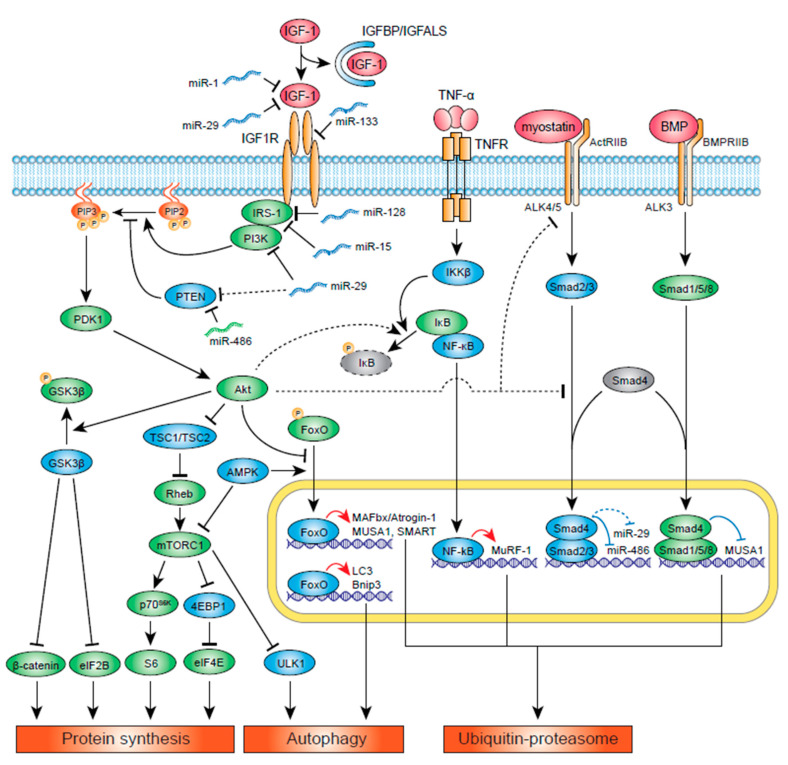
IGF-1 signaling pathways. In the figure, the signaling molecules and miRNAs that activate protein synthesis and/or inhibit protein degradation are shown in green, while the ones that inhibit protein synthesis and/or activate protein degradation are shown in blue. The majority of IGF-1 in the body are bound to IGFBP and IGFALS, and its activity is suppressed. Once IGF-1 binds to IGF-1R, IRS-1 and PI3K are recruited and activated. PI3K converts PIP2 to PIP3, which activates PDK1 and Akt. Akt activates protein synthesis via activation of ribosomal protein S6 and the translation initiation factor eIF4E downstream of mTORC1, and activation of β-catenin and eIF2B downstream of GSK3β. Akt can suppress UPS activity via inhibition of FoxO-mediated transcription of E3 ubiquitin ligases MAFbx/Atrogin-1, MUSA1, and SMART. MuRF1 expression is induced by cytokines such as TNF-α via NF-κB pathway. Akt could phosphorylate IκΒ and activate the NF-κΒ pathway, although it has not been shown in skeletal muscle and multiple studies have shown IGF-1 activation does not alter MuRF1 expression. Myostatin and BMP signaling compete against each other for their usage of Smad4. Activation of myostatin inhibits BMP-mediated Smad1/5/8 translocation to the nucleus, thus inhibiting MUSA1-mediated UPS activity. Akt can also downregulate ActRIIB and inhibit ALK4/5 via unknown mechanisms. Although it has not been shown in skeletal muscle, Akt can interact directly with unphosphorylated Smad3 to sequester it outside the nucleus. Several miRNAs have been shown to regulate IGF-1 signaling. miR-486 is inhibited by the myostatin/Smad pathway, resulting in inhibition of IGF-1 signaling via PTEN increase. miR-1 and miR-133 target IGF-1 and IGF-1R, respectively, and their expression is reduced during muscle hypertrophy. IRS-1 could be inhibited by miR-128 and miR-15. LncIRS1 (not shown in the figure), which is upregulated in hypertrophic muscles, can act as sponge for miR-15, resulting in activation of IRS-1. Note that studies have shown conflicting evidence on miR-29′s role in IGF-1 signaling in skeletal muscle, and it may potentiate or inhibit IGF-1 signaling. Pathways that are unclear and/or not shown in skeletal muscle are shown in dotted lines.

**Figure 2 cells-09-01970-f002:**
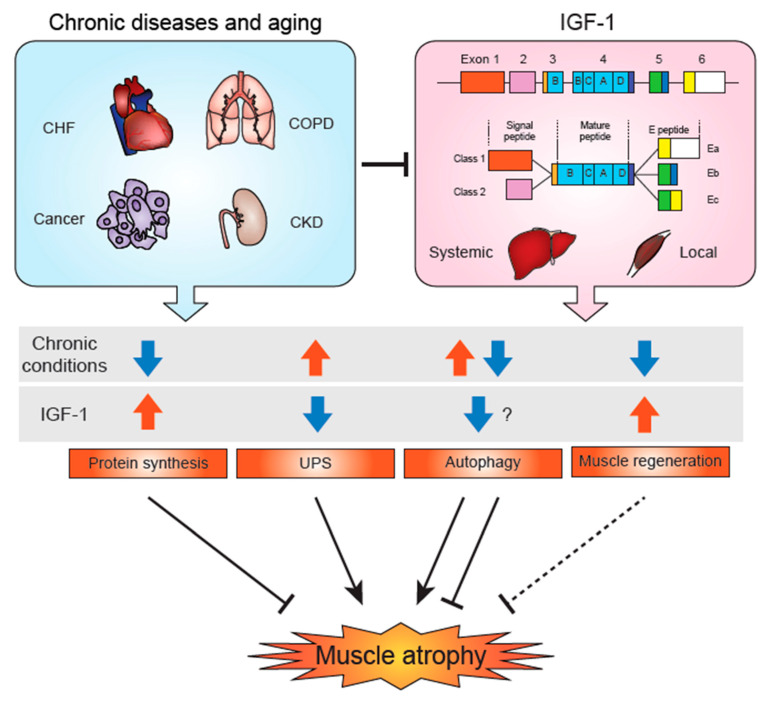
IGF-1 and skeletal muscle atrophy in chronic diseases and aging. In various chronic disease conditions, such as congestive heart failure (CHF), cancer, chronic obstructive pulmonary disease (COPD), and chronic kidney disease (CKD), and aging, muscle atrophy develops through various mechanisms: decreased protein synthesis, increased UPS, and lowered muscle regeneration. Depending on the pathophysiological conditions, autophagy could be increased or decreased, and both excessive and defective autophagy could lead to muscle atrophy. IGF-1 is thought to decrease autophagy, but the role of IGF-1 regulation of autophagy in chronic disease-induced muscle atrophy is yet to be determined. IGF-1 stimulates skeletal muscle regeneration via activation of satellite cells. Systemic (circulating) IGF-1 is predominantly produced in the liver, whereas locally produced IGF-1 likely acts in a paracrine or autocrine manner. The first two exons of IGF-1 are mutually exclusive and generate different signal peptides, termed Class 1 (exon 1) and Class 2 (exon 2). The mature IGF-1 peptide is coded in exons 3 and 4 (B, C, A, and D domains). Three types of C-terminus E-peptides are generated by alternative splicing. Ea is from exon 6, Eb is from exon 5, and Ec is from part of exons 5 and 6. Class 2 IGF-1 is mainly expressed in the liver (considered to be the systemic isoform), and Class 1 IGF-1 is mainly expressed in peripheral tissues including skeletal muscle. Both systemic and local IGF-1 levels are decreased in various chronic disease conditions, and the combination of these reductions affect protein synthesis, UPS activity, autophagy, and muscle regeneration and regulate the development of muscle atrophy.

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
