# Peer review of "Mechanisms of IGF-1-Mediated Regulation of Skeletal Muscle Hypertrophy and Atrophy"

_cells, 2020, doi:10.3390/cells9091970_

Round 1

Reviewer 1 Report

In this review the authors provide a comprehensive overview of the role of IGF-1 in the regulation of skeletal muscle hypertrophy/ atrophy under different physiological and pathological conditions. The review describes recent advances on the signaling pathways regulating skeletal muscle hypertrophy/ atrophy.

In general, the focus of the review is supposed to be IGF-1. At the same time, large portions of this manuscript describe general details of UPS, autophagy, micro RNAs and long noncoding RNAs, and other processes with very little information on how they are related/connected to IGF-1. There are numerous recent and excellent reviews on UPS and autophagy. There is no need to describe these details in the current manuscript. Focus instead on the IGF-1 and its role in skeletal muscle hypertrophy/ atrophy. The review can be shortened at least in half when unnecessary information is removed. Because many parts of the review are so vague it is difficult to read and comprehend.

I have a few other suggestions/questions for the authors:

Abstract is too long and unfocused. It lists too many details that will be described later in the body of the manuscript. Please revise to make it straightforward and easy to read.

Fix:  “can phosphorylates and inhibits”

Introduction starts with IGF-1 in animal models then switches abruptly to healthy young subjects. Please revise to make the transitions smoother and make introduction more straightforward.

Figure 1. “The majority of IGF-1 exists” Exists where?

A lot of abbreviations without descriptions.

Figure description text is too long. Please revise to make it shorter and more focused.

Muscle protein synthesis and IGF-1 signaling

Muscle protein synthesis: Myostatin

These chapters list a large number of known from the literature data without providing logical connections, transitions, and interpretations. Please revise to make it shorter, clearer, focused, and reader friendly.

Muscle protein degradation: UPS

More than two pages of the description of the details of UPS with very little information on the connection to IGF-1. Shorten description of UPS and focus on the connection and interpretation/discussion of the connection to IGF-1.

Muscle protein degradation: Autophagy

More than one and a half pages of the description of the details of autophagy with very little information on IGF-1. At the first page IGF-1 is not even mentioned. The rest of the text has very little information on IGF-1.

Figure 2. Description is too long. Please revise to make it shorter and more focused.

IGF-1, satellite cells and skeletal muscle regeneration

“IGF-1 overexpression in differentiated L6E9 cells” Not everybody knows that L6 is rat myoblast cell line. Please describe clearer.

The rest of the text: Please make it shorter, clear, focused, and more reader friendly.

Conclusion. Please make it shorter, clear, and focused.

Author Response

> In this review the authors provide a comprehensive overview of the role of IGF-1 in the regulation of skeletal muscle hypertrophy/ atrophy under different physiological and pathological conditions. The review describes recent advances on the signaling pathways regulating skeletal muscle hypertrophy/ atrophy.

> In general, the focus of the review is supposed to be IGF-1. At the same time, large portions of this manuscript describe general details of UPS, autophagy, micro RNAs and long noncoding RNAs, and other processes with very little information on how they are related/connected to IGF-1. There are numerous recent and excellent reviews on UPS and autophagy. There is no need to describe these details in the current manuscript. Focus instead on the IGF-1 and its role in skeletal muscle hypertrophy/ atrophy. The review can be shortened at least in half when unnecessary information is removed. Because many parts of the review are so vague it is difficult to read and comprehend.

The entire manuscript is revised to make it to be more focused on IGF-1, and some detailed information that are available elsewhere are consolidated to make it easier to read and to help the reader more focused on the topic.

> I have a few other suggestions/questions for the authors:
> Abstract is too long and unfocused. It lists too many details that will be described later in the body of the manuscript. Please revise to make it straightforward and easy to read.

Abstract is significantly shortened, more focused and straightforward.

> Fix: “can phosphorylates and inhibits”
This part is removed as the entire abstract has been revised.

> Introduction starts with IGF-1 in animal models then switches abruptly to healthy young subjects. Please revise to make the transitions smoother and make introduction more straightforward.
Modified to start with: "Studies in various models in cell culture, animals and humans have evaluated...". Also included a new reference.

> Figure 1. “The majority of IGF-1 exists” Exists where?
Fixed to: "The majority of IGF-1 in the body are bound to IGFBP and ALS..."

> A lot of abbreviations without descriptions.

Non-abbreviated words are included where the words are first used in the manuscript.

> Muscle protein synthesis and IGF-1 signaling
> Muscle protein synthesis: Myostatin
> These chapters list a large number of known from the literature data without providing logical connections, transitions, and interpretations. Please revise to make it shorter, clearer, focused, and reader friendly.

These sections have been revised according to the reviewer's suggestions, and made shorter and more focused.

> Muscle protein degradation: UPS
> More than two pages of the description of the details of UPS with very little information on the connection to IGF-1. Shorten description of UPS and focus on the connection and interpretation/discussion of the connection to IGF-1.

Our intention was to describe the overlap between IGF-1 signaling and regulatory mechanisms of E3 ubiqutin ligases. This point was made clear in this section, and the discussion is made more focused around IGF-1.

> Muscle protein degradation: Autophagy
> More than one and a half pages of the description of the details of autophagy with very little information on IGF-1. At the first page IGF-1 is not even mentioned. The rest of the text has very little information on IGF-1.

As suggested by the reviewer, the first introductory part of autophagy has been significantly shortened, and the entire section is made more focused on IGF-1.

> Figure description text is too long. Please revise to make it shorter and more focused.
> Figure 2. Description is too long. Please revise to make it shorter and more focused.

The figure legends of both Fig. 1 and 2 are shortened and more focused.

> IGF-1, satellite cells and skeletal muscle regeneration
> “IGF-1 overexpression in differentiated L6E9 cells” Not everybody knows that L6 is rat myoblast cell line. Please describe clearer.
> The rest of the text: Please make it shorter, clear, focused, and more reader friendly.
The description of L6E9 and L6A1 has been added. The entire section is shortened and made more focused on IGF-1.

> Conclusion. Please make it shorter, clear, and focused.

As described above, the entire manuscript, especially where pointed out by the reviewer, are made shortened and more focused.

Reviewer 2 Report

The study deals with mechanisms behind the balance between protein synthesis and degradation in skeletal muscle, and in that context the clinically important problems atrophy and sarcopenia. The focus is on the significance of the positive influence of IGF11 on protein synthesis but also compounds and pathways leading to degradation of protein or inhibition of synthesis are dealt with. The topic, particular IGF1, are systematically looked upon from many sides, and the review presents and discusses detailed the various aspects.

Considering the importance of the balance between synthesis and degradation, I would be helpful for the reader if more of the presented information on interaction between the IGF1 and the “negative” pathways could be visualized in a figure like fig 1

In the introduction, the terms muscle cell and myocyte is used. These are normally applied to the cells in cardiac and smooth muscle cells, while myofiber or muscle fiber are used in skeletal muscle.

In the section : Muscle protein synthesis and IGF1 signaling, the critical role of mTOR/availability of amino acids for the synthesis of nucleic acid, glucose and ATP should be reconsidered, e.g. glucose is build in plants and algae.

Author Response

> The study deals with mechanisms behind the balance between protein synthesis and degradation in skeletal muscle, and in that context the clinically important problems atrophy and sarcopenia. The focus is on the significance of the positive influence of IGF11 on protein synthesis but also compounds and pathways leading to degradation of protein or inhibition of synthesis are dealt with. The topic, particular IGF1, are systematically looked upon from many sides, and the review presents and discusses detailed the various aspects.

> Considering the importance of the balance between synthesis and degradation, I would be helpful for the reader if more of the presented information on interaction between the IGF1 and the “negative” pathways could be visualized in a figure like fig 1

In the new Fig. 1, the signaling molecules and miRNAs that activate protein synthesis and/or inhibit protein degradation ("positive" regulators) are shown in green, and the ones that inhibit protein synthesis and/or activate protein degradation ("negative" regulators) are shown in blue.

> In the introduction, the terms muscle cell and myocyte is used. These are normally applied to the cells in cardiac and smooth muscle cells, while myofiber or muscle fiber are used in skeletal muscle.

Myocyte is replaced with myofiber throughout the manuscript.

> In the section : Muscle protein synthesis and IGF1 signaling, the critical role of mTOR/availability of amino acids for the synthesis of nucleic acid, glucose and ATP should be reconsidered, e.g. glucose is build in plants and algae.

The sentences are revised as: Mammalian target of rapamycin (mTOR) is a downstream target of Akt, and in mammalian cells mTOR activity is tightly regulated by amino acid availability to the cells. As amino acids are necessary to build proteins, nucleic acid, glucose and ATP in the body, mTOR activity is highly correlated with the anabolic/catabolic balance.

Reviewer 3 Report

The article by Yoshida et al aims to present the mechanism of IGF-1 dependent regulation of skeletal muscle homeostasis. The authors deeply investigate the role of IGF-1 in muscle protein synthesis and degradation focusing on the UPS and autophagyc pathways. Moreover, the authors review the alternative splicing of IGF-1 mRNAs to produce local and systemic forms and describe the regulation of IGF-1 signaling by atrophy-related miRNAs.

The review is well written and structured, however some more details need to be added:

Minor

-Muscle homeostasis results from the delicate balance between catabolic and anabolic processes both regulated by energy expenditure. AMPK is an energy sensor kinase and its activity is strictly correlated to IGF-1 signaling, since AMPK inhibits mTOR and stimulates FoxO dependent protein metabolism. The authors should better discuss this point to perform a more comprehensive analysis of IGF-1 involvement in muscle metabolic homeostasis.

-In the paragraph “IGF-1 changes in chronic conditions” the authors investigate the effect of circulating and muscle specific IGF-1 in several chronic conditions. Although they performed a deep investigation about cancer and cancer cachexia, they just mentioned the role of IGF-1 in other myopathies or neuromuscular disorders. The author should deeper review this point  in order to present a complete overview of IGF-1 regulation in chronic diseases.

Further in the paragraph “IGF-1 and sarcopenia” the authors focus on the description of aging related sarcopenia. Of course sarcopenia refers to age-related loss of muscle mass and contractile properties, however more recently the sarcopenic condition has been related to chronic disorders such as obesity and liver diseases. Therefore, authors should describe the role of IGF-1 in more than one sarcopenic conditions or limit paragraph title to aging-related sarcopenia.

Author Response

> The article by Yoshida et al aims to present the mechanism of IGF-1 dependent regulation of skeletal muscle homeostasis. The authors deeply investigate the role of IGF-1 in muscle protein synthesis and degradation focusing on the UPS and autophagyc pathways. Moreover, the authors review the alternative splicing of IGF-1 mRNAs to produce local and systemic forms and describe the regulation of IGF-1 signaling by atrophy-related miRNAs.

> The review is well written and structured, however some more details need to be added:

## Minor
> Muscle homeostasis results from the delicate balance between catabolic and anabolic processes both regulated by energy expenditure. AMPK is an energy sensor kinase and its activity is strictly correlated to IGF-1 signaling, since AMPK inhibits mTOR and stimulates FoxO dependent protein metabolism. The authors should better discuss this point to perform a more comprehensive analysis of IGF-1 involvement in muscle metabolic homeostasis.

A new section "Muscle energy homeostasis: AMPK and IGF-1" has been included, and AMPK signaling is added to Fig. 1.

> In the paragraph “IGF-1 changes in chronic conditions” the authors investigate the effect of circulating and muscle specific IGF-1 in several chronic conditions. Although they performed a deep investigation about cancer and cancer cachexia, they just mentioned the role of IGF-1 in other myopathies or neuromuscular disorders. The author should deeper review this point in order to present a complete overview of IGF-1 regulation in chronic diseases.

> Further in the paragraph “IGF-1 and sarcopenia” the authors focus on the description of aging related sarcopenia. Of course sarcopenia refers to age-related loss of muscle mass and contractile properties, however more recently the sarcopenic condition has been related to chronic disorders such as obesity and liver diseases. Therefore, authors should describe the role of IGF-1 in more than one sarcopenic conditions or limit paragraph title to aging-related sarcopenia.

The section "IGF-1 changes in chronic conditions" is combined with “IGF-1 and sarcopenia" and title is changed to "IGF-1 changes in chronic conditions and aging-associated sarcopenia". More discussion is included for chronic disease-associated muscle atrophy other than cancer.

Round 2

Reviewer 1 Report

The authors have significantly modified the manuscript. These modifications made it more focused and easier to read.

I have a few suggestions for the authors:

Line 36: “a balance between new protein synthesis and old protein degradation”. I suggest: “a balance between synthesis of new proteins and degradation of old proteins”.

Line 46: Needs a dot after IGF-1 signaling pathways.

Line 122: it remains to be determined whether the same mechanism

Line 125: mass.4.

Line 143: wheres MuRF1 activation

Line 206: starvation.Various skeletal muscle

Line 253: autophagy markesr

Line 321: pro-IGF-1 has more potent to activate IGF-1R

Line 468: muscular dystrophies (CMD, FSHD) and ALS. Define ALS here since this abbreviation was used before for the acid labile subunit (ALS).

Line 547: mouse models of amyotrophic lateral sclerosis (ALS). ALS abbreviation is used twice to refer to two completely different things.

Line 508: Furthermore

Line 533: By RNA sequencing of hypertrophic and leaner broilers, Li et al. identified a novel lncRNA, termed lncIRS1, is upregulated in hypertrophic muscles.

Line 627: and these pathways likely interact with each other

Author Response

> Line 36: “a balance between new protein synthesis and old protein degradation”. I suggest: “a balance between synthesis of new proteins and degradation of old proteins”.

> Line 46: Needs a dot after IGF-1 signaling pathways.
> Line 143: wheres MuRF1 activation
> Line 206: starvation.Various skeletal muscle
> Line 253: autophagy markesr
> Line 321: pro-IGF-1 has more potent to activate IGF-1R
> Line 508: Furthermore
> Line 533: By RNA sequencing of hypertrophic and leaner broilers, Li et al. identified a novel lncRNA, termed lncIRS1, is upregulated in hypertrophic muscles.
> Line 627: and these pathways likely interact with each other
> Line 122: it remains to be determined whether the same mechanism

Above suggested corrections are accepted and all the typos are corrected.

> Line 125: mass.4.
We suppose this is an editorial error. It is not in the submitted file.

> Line 468: muscular dystrophies (CMD, FSHD) and ALS. Define ALS here since this abbreviation was used before for the acid labile subunit (ALS).
> Line 547: mouse models of amyotrophic lateral sclerosis (ALS). ALS abbreviation is used twice to refer to two completely different things.

To avoid any confusion, acid labile complex is replaced with its formal name IGF binding protein acid labile complex (IGFALS) throughout the manuscript and figure 1.